# An Ethyl-Thioglycolate-Functionalized Fe_3_O_4_@ZnS Magnetic Fluorescent Nanoprobe for the Detection of Ag^+^ and Its Applications in Real Water Solutions

**DOI:** 10.3390/nano13131992

**Published:** 2023-07-01

**Authors:** Xin Chen, Jie Chen, Mingshuo Ma, Shihua Yu, Zhigang Liu, Xiaodan Zeng

**Affiliations:** 1School of Chemical and Pharmaceutical Engineering, Jilin Institute of Chemical Technology, Jilin 132022, China; chenxin92@jlict.edu.cn (X.C.); ysh@jlict.edu.cn (S.Y.); 2Center of Characterization and Analysis, Jilin Institute of Chemical Technology, Jilin 132022, China; jiechendr@163.com (J.C.); naya221@163.com (M.M.)

**Keywords:** zinc sulfide, Fe_3_O_4_, nanoparticle, fluorescence

## Abstract

Ethyl-thioglycolate-modified Fe_3_O_4_@ZnS nanoparticles (Fe_3_O_4_@ZnS-SH) were successfully prepared using a simple chemical precipitation method. The introduction of ethyl thioglycolate better regulated the surface distribution of ZnS, which can act as a recognition group and can cause a considerable quenching of the fluorescence intensity of the magnetic fluorescent nanoprobe, Fe_3_O_4_@ZnS-SH. Benefiting from stable fluorescence emission, the magnetic fluorescent nanoprobe showed a highly selective fluorescent response to Ag+ in the range of 0–400 μM, with a low detection limit of 0.20 μM. The magnetic fluorescent nanoprobe was used to determine the content of Ag^+^ in real samples. A simple and environmentally friendly approach was proposed to simultaneously achieve the enrichment, detection, and separation of Ag^+^ and the magnetic fluorescent nanoprobe from an aqueous solution. These results may lead to a wider range of application prospects of Fe_3_O_4_ nanomaterials as base materials for fluorescence detection in the environment.

## 1. Introduction

Silver, an important transition metal, is of great commercial value and has marked antibacterial properties, meaning that it is usually used in high-strength and high-corrosion-resistance alloys and jewelry, dental and pharmaceutical preparations, and implanted prostheses [1,2]. As an important noble metal, silver is also widely used in the electronics industry as well as electroplating. In addition, soluble silver compounds and colloidal silver are usually used for treating some diseases, i.e., gastroenteritis and infections, because of their antibacterial activity. The silver content in environmental matrices is increasing with the widespread use of silver-containing compounds in procedures in medicine, industry, and so on [3].

On the other hand, silver ions (Ag^+^) are one of the most toxic heavy metals [4]. They not only cause irreversible damage to the human body, animals, and plants, but also cause great harm to water and the environment. The limit for Ag^+^ in drinking water is 0.1 mg/L or 0.93 μM based on the U.S. EPA [5]. Therefore, monitoring and detecting Ag^+^ are of great significance in environmental water, food, agricultural products, and so on.

Well-known and commonly used methods for trace Ag^+^ analysis include inductively coupled plasma mass spectrometry [6], electrochemiluminescence [7], flow injection–flame atomic absorption spectrometry [8], ICP-MS [9], the electrochemical method [10], and laser ablation microwave plasma torch optical emission spectrometry [11]. These methods can offer wide linearity ranges, low detection limits, and excellent precision. However, these methods are often easily affected by the introduction of sample solutions, leading to low measurement sensitivity and result accuracy.

The fluorescence technique [12,13,14] has been an attractive research tool due to its high selectivity and sensitivity, straightforward equipment requirements, and ease of operation. Many fluorescent probes have been developed, including small organic molecules [15,16,17], noble metal nanoclusters [18,19,20], quantum dots [21,22,23], and carbon nanodots [24,25]. In particular, quantum dots attract significant attention from chemists as a new class of inorganic fluorophores due to their unique superior properties compared to classical organic fluorophores [26,27,28]. Functionalization is an effective method for improving fluorescence selectivity and sensitivity. Several types of quantum dots used as sensor probes for the detection of Ag^+^ have been reported [29,30,31]. However, the reported probes were difficult to separate from the environment completely or worked via the fluorescence-quenching approach, which was frequently caused by many possible species—especially heavy metal ions—which leads to low selectivity, introduces pollution, and hinders their application in actual environmental detection. In addition, the enrichment and separation of a fluorescent probe and target are still difficult. Therefore, it still remains challenging to develop a facile method to detect heavy ions with high fluorescence sensitivity and less pollution.

Fe_3_O_4_ is a class of nanoparticle that has been widely studied for its super paramagnetic properties, high surface area, easy preparation and separation, high adsorption ability, and biocompatibility [32,33,34,35,36]. More importantly, its ease of function features may provide it with promising applications in fluorescence detection [37,38,39]. From the discussion above, the introduction of Fe_3_O_4_ into a nanostructure is expected to provide better enrichment, detection, and separation effects and higher sensitivity. Therefore, the magnetic technique combined with fluorescent molecules can overcome the disadvantages described above.

To address the above challenges, based on the strong coordination capacity of ethyl thioglycolate groups with Ag^+^, a novel magnetic fluorescence nanoprobe (Fe_3_O_4_@ZnS-SH, Figure 1a) was prepared through a simple chemical precipitation process. Because the fluorescence intensity of the magnetic fluorescence nanoprobe was remarkably quenched by the silver ions, a fluorescence method for the determination of Ag^+^ was established. In addition, the magnetic fluorescence nanoprobe was used for the determination of Ag^+^ water samples successfully. The magnetic fluorescence nanoprobe combines the advantages of fluorescence analysis and magnetic separation (Figure 1b). It provided a simple, highly selective, and less polluting method to determine Ag^+^.

## 2. Experimental

### 2.1. Materials and Reagents

Iron (III) chloride hexahydrate (99%), iron dichloride (99%), sodium acetate trihydrate (99%), zinc acetate (99%), sodium sulfide (99%), polyethylene glycol 2000, and ethyl thioglycolate (97%) were purchased from Sigma Aldrich.

### 2.2. Physical Characterizations

Scanning electron microscope (SEM) images were captured by JSM6490LV (JEOL, Tokyo, Japan). Transmission electron microscopy (TEM) images were captured by JEM 2100 (JEOL, Tokyo, Japan). Fourier transform infrared (FTIR) spectroscopy was carried out using Thermo Scientific Nicolet IS50 (Thermofisher, Waltham, MA, USA). X-ray diffraction (XRD) was performed using on D8FOCUS (Bruker, Karlsruhe, German). X-ray photoelectron spectra (XPS) were obtained using Krayos AXIS Ultra DLDX (Shimadzu, Kyoto, Japan). The magnetic behavior (VSM) was recorded on dried samples by using a 7404 vibrating sample magnetometer (LakeShore, Carson, CA, USA). Thermal gravimetric analysis (TGA) was carried out on a Discovery SDT650 synchronous TGA-DTA instrument (TA, Los Angeles, CA, USA) heated from room temperature to 800 °C (10 °C/min) in nitrogen atmosphere. Lifetime measurements were carried out on FluoroMax-plus instrument (Horiba, Irvine, CA, USA). All fluorescence measurements were performed on a Cary Eclipse spectrophotometer (Varian, Palo Alto, CA, USA). The excitation wavelength was 370 nm and the excitation and emission slit width was set as 10/10 nm.

### 2.3. Synthesis of Fe_3_O_4_ Magnetic Nanoparticles

Fe_3_O_4_ magnetic nanoparticles were synthesized by means of the hydrothermal method reported in the literature. Ferric chloride hexahydrate (2.7 g) was first dissolved with ethanediol (60 mL) in a beaker (100 mL, 85 °C water bath). The mixed solution was stirred, and a black solution was obtained. Then, CH_3_COONa (7.2 g) was added to the solution sequentially, the mixed solution was stirred for 30 min and transferred to stainless steel autoclave that lined with polytetrluoroethylene (100 mL, 200 °C, 12 h). The autoclave was cooled to 25 °C after the reaction. The black products were collected, washed with ethanol for 3 times, and dried at 80 °C for 6 h.

### 2.4. Synthesis of Fe_3_O_4_@ZnS Magnetic Nanoparticles

Fe_3_O_4_ magnetic nanoparticles (0.3 g) were dispersed in deionized water and ammonia (5 mL) to the final volume of 100 mL. The solution was stirred for 30 min in the water bath (80 °C). Then, the mixed solution was injected with the solutions of Zn(Ac)_2_·2H_2_O (2.5 mmol, 50 mL) and Na_2_S·9H_2_O (2.5 mmol, 50 mL) and continuously stirred for 6 h in a water bath (80 °C). After the reaction, the products were washed with water several times and further dried at 80 °C for 6 h. Fe_3_O_4_@ZnS nanoparticles were eventually obtained.

### 2.5. Synthesis of Fe_3_O_4_@ZnS@Ethyl Thioglycolate Magnetic Nanoparticles

Fe_3_O_4_@ZnS microspheres (30 mg) were dispersed in a mixed solution system (1 mL chloroform, 1 mL water and 1 mL ethyl thioglycolate) and the mixture was left until it was divided into two layers of solution. The upper solution was removed and was added with alcohol (1 mL) to deposit the Fe_3_O_4_@ZnS@Ethyl thioglycolate nanoparticles. The product was treated with a centrifuge for 30 min (8000 rpm) and added to water (1 mL) and NaOH, and then was filtered by filter membrane. Alcohol (1 mL) was added to the filtered solution and centrifuged for 30 min (8000 rpm). Finally, the product was dried at 80 °C for 6 h. Fe_3_O_4_@ZnS@ Ethyl thioglycolate (Fe_3_O_4_@ZnS-SH) nanoparticles were eventually obtained.

### 2.6. Determination of Ag^+^ by Magnetic Fluorescence Nanoprobe Fe_3_O_4_@ZnS-SH

Before the spectroscopic measurements, the working solutions of the magnetic fluorescence nanoprobe Fe_3_O_4_@ZnS-SH (1 mg/mL) and different concentrations of AgNO_3_ (0–400 µM) were prepared by diluting the high concentration solution to the required concentration. To investigate the selectivity of the magnetic fluorescence nanoprobe Fe_3_O_4_@ZnS-SH toward Ag^+^, the following metal ion salts Co^2+^, Ni^2+^, Al^3+^, Cu^2+^, Zn^2+^, Cd^2+^, Fe^3+^, Fe^2+^, K^+^, Ca^2+^, Na^+^, Pb^2+^, and Hg^2+^ were used. All spectra were collected three times to eliminate the error induced by the suspension change and instrument fluctuation.

## 3. Results and Discussion

### 3.1. SEM and TEM Analysis

SEM and TEM analyses of Fe_3_O_4_, Fe_3_O_4_@ZnS, and Fe_3_O_4_@ZnS@Ethyl thioglycolate magnetic nanoparticles were conducted to provide the morphological changes of the samples more directly and intuitively during synthesis. The SEM and TEM images of Fe_3_O_4_, Fe_3_O_4_@ZnS, and Fe_3_O_4_@ZnS@Ethyl thioglycolate magnetic nanoparticles are shown in Figure 2. As shown in Figure 2a,d, the surface of Fe_3_O_4_ was relatively smooth and flat. Additionally, a thin ZnS layer was formed on the surface when Fe_3_O_4_ nanoparticles were coated with ZnS (Figure 2b,e), leading to the particle diameter becoming larger and the dispersity being improved. The smaller-diameter particles that were scattered around were some residual ZnS quantum dots. After the reaction of ethyl thioglycolate on the surface of Fe_3_O_4_@ZnS (Figure 2c,f), the surface of the Fe_3_O_4_@ZnS@Ethyl thioglycolate magnetic nanoparticles’ structure was more uniform and smoother than Fe_3_O_4_@ZnS, and the volume increased, confirming the successful synthesis of Fe_3_O_4_@ZnS@Ethyl thioglycolate.

### 3.2. Characterization of Fe_3_O_4_@ZnS@Ethyl Thioglycolate Magnetic Nanoparticles

First, the FT-IR spectra of Fe_3_O_4_, Fe_3_O_4_@ZnS, Fe_3_O_4_@ZnS@Ethyl thioglycolate (Fe_3_O_4_@ZnS-SH) and ethyl thioglycolate are shown in Figure 3. As is shown in the spectra of Fe3O4, the absorption peaks at 3436 cm^−1^ and 1562 cm^−1^ belong to -OH and C=C, respectively [40]. The strong characteristic absorption peak at 584 cm^−1^ is due to the stretching vibration of Fe-O. The peak at 1420 cm^−1^ was attributed to C-O tensile vibration, and the peak at 1652 cm^−1^ may be due to existing generated carbonyl group from the transformation of some parts of polyethylene glycol, which indicated the presence of surfactant in the product. As for the spectra of Fe_3_O_4_@ZnS, the absorption peaks at 580 cm^−1^ and 627 cm^−1^ represent Fe-O and Zn-S, respectively. The -SH bond stretching vibration peaks from ethyl thioglycolate are located at 2600–2500 cm^−1^ and disappeared in the spectra of Fe_3_O_4_@ZnS@Ethyl thioglycolate. Compared with the FT-IR spectroscopy of ethyl thioglycolate, the disappearance of the characteristic absorption peak of the -SH stretching vibration in Fe_3_O_4_@ZnS@Ethyl thioglycolate also confirmed the successful combination of the ethyl thioglycolate and Fe_3_O_4_@ZnS. These results prove that the Fe_3_O_4_@ZnS@Ethyl thioglycolate magnetic nanoparticles were successful synthesized.

Secondly, XRD technology was applied to analyze the structural patterns of ZnS, Fe_3_O_4_, Fe_3_O_4_@ZnS and Fe_3_O_4_@ZnS@Ethyl thioglycolate nanoparticles (Figure 4) and the XRD diffraction patterns of the ZnS, Fe_3_O_4_, Fe_3_O_4_@ZnS and Fe_3_O_4_@ZnS@Ethyl thioglycolate structure were compared. In Figure 4, the data for ZnS nanoparticles showed the diffraction peaks at 2θ = 28.9°, 47.7°, and 57.0°, which can be indexed to the (111), (220), and (311) planes of ZnS, respectively. Additionally, the data for Fe_3_O_4_ nanoparticles showed the diffraction peaks at 2θ = 30.1°, 35.6°, 43.1°, 57.1°, and 62.8°, which can be indexed to the (220), (311), (222), (511), and (440) planes of Fe_3_O_4_, respectively [41]. In addition, these diffraction peaks did not change in the XRD patterns of Fe_3_O_4_@ZnS and Fe_3_O_4_@ZnS-SH. The surface modification of the magnetic nanoparticles introduced no change in the crystal structure and the data indicated the stability of the Fe_3_O_4_. Thereby, the results demonstrated that the ZnS and ethyl thioglycolate were successfully coated onto the surface of the Fe_3_O_4_ nanoparticles.

Next, the results of elemental analysis using XPS in Figure 5a confirmed the difference between Fe_3_O_4_, Fe_3_O_4_@ZnS and Fe_3_O_4_@ZnS-SH. The peaks of Fe2p (725 eV, 708 eV), O1s (530 eV), and C1s (283 eV) that appeared in Fe_3_O_4_@ZnS@Ethyl thioglycolate were also observed in Fe_3_O_4_@ZnS and Fe_3_O_4_, while additional peaks of Zn2p (1019 eV, 1042 eV) and S2p (159 eV) were also observed in Fe_3_O_4_@ZnS and Fe_3_O_4_@ZnS-SH, which were seen more clearly in the XPS Zn2p and XPS S2p shown in Figure 5c,d [42]. Due to the core–shell structure of nanoparticles, the intensities of the outer shell component lines of O and C decreased in turn and the intensities of the component lines of Zn and S decreased in turn.

The characteristic Fe2p lines of Fe_3_O_4_ are presented in Figure 5b, positioned at 709.01 eV and 722.12 eV in the case of Fe^2+^ and at 713.95 eV and 724.95 eV for Fe^3+^, respectively. In addition, in the deconvolution of the Zn2p doublet spectrum (Figure 5c), two components were observed. The peaks positioned a 1019.02 eV and 1042.03 eV and 1021.51 eV and 1044.41 eV were ascribed to 2p (3/2) and 2p (1/2), respectively. Meanwhile, the peaks that were positioned at 159.35 eV and 161.47 eV (Figure 5d) were assigned to metal sulfide S^2−^ (2p3/2 and S 2p1/2) from ZnS, respectively [43]. Therefore, the above results proved the successful modification of ethyl thioglycolate on Fe_3_O_4_@ZnS.

Then, the magnetic hysteresis (M(H)) loops of three nanoparticles (Fe_3_O_4_, Fe_3_O_4_@ZnS and Fe_3_O_4_@ZnS@Ethyl thioglycolate) were measured in external fields between -20 kG and 20 kG (Figure 6). The results showed that these three nanoparticles had similar M (H) loops with almost complete reversibility, except for the fact that there were small distinctions in the magnetization saturation values. The magnetization saturation value of the Fe_3_O_4_@ZnS (33 emu/g) was lower than that of the Fe_3_O_4_ nanoparticles (63 emu/g) because the introduction of the ZnS enhanced the weight of the nanoparticles [44,45]. Similarly, the magnetization saturation value of Fe_3_O_4_@ZnS@Ethyl thioglycolate (25 emu/g) was the lowest because of the effect of the outer ZnS layer and ethyl thioglycolate. The magnetism of nanomaterials is an important parameter for magnetic materials if they are to be used in rapid separation techniques. So, the excellent reversibility and magnetization saturation value of the Fe_3_O_4_@ZnS@Ethyl thioglycolate nanoparticles implied that this nanoparticle may be perfect for use to identify heavy metals and separate them from the test environment for recovery. In addition, Figure 6 (inset figure) shows that the Fe_3_O_4_@ZnS@Ethyl thioglycolate nanoparticles can be easily dragged to the side using an external magnet, and the solution became transparent immediately, demonstrating that the Fe_3_O_4_@ZnS@Ethyl thioglycolate nanoparticles possess the favorable magnetic ability required to realize the targeted identification and separation.

Finally, to further investigate the characteristics of the synthesized nanoparticles, three samples were burned at temperatures up to 800 ℃ for TGA analysis in a nitrogen atmosphere. Figure 7 shows the thermogravimetric analysis results for Fe_3_O_4_, Fe_3_O_4_@ZnS and Fe_3_O_4_@ZnS@Ethyl thioglycolate. As shown, the Fe_3_O_4_ magnetic nanoparticles exhibited excellent thermal stability due to lacking a surface coating, and the mass loss was 8.71%. When Fe_3_O_4_ magnetic nanoparticles were coated with ZnS, the Fe_3_O_4_@ZnS microspheres had a mass loss of 12% due to the ZnS layer on the surface. The weight loss of Fe_3_O_4_@ZnS@Ethyl thioglycolate was 22% at 650 °C, and the maximum total loss was reached at 650 °C. Within the range of 400–700 °C, Fe_3_O_4_@ZnS@Ethyl thioglycolate presented significantly more mass loss than Fe_3_O_4_ and Fe_3_O_4_@ZnS, ascribed to the further destruction of ethyl thioglycolate. The above results indicate the successful synthesis of ethyl thioglycolate on the surface of Fe_3_O_4_@ZnS.

Combined with the above data, it can be proved that the Fe_3_O_4_@ZnS@Ethyl thioglycolate magnetic nanoparticles (Fe_3_O_4_@ZnS-SH) were successfully synthesized.

### 3.3. Performance Analysis of Magnetic Fluorescence Nanoprobe Fe_3_O_4_@ZnS-SH

Figure 8 showed the comparison of fluorescence characteristic spectra of Fe_3_O_4_@ZnS and Fe_3_O_4_@ZnS-SH before and after adding Ag^+^, respectively. It can be seen from Figure 8 that the fluorescence intensity Fe_3_O_4_@ZnS is slight stronger than that of Fe_3_O_4_@ZnS-SH. The addition of Ag^+^ can quench both Fe_3_O_4_@ZnS and Fe_3_O_4_@ZnS-SH. However, it was obvious that the quenching degree of Ag^+^ on Fe_3_O_4_@ZnS-SH is larger than Ag^+^ on Fe_3_O_4_@ZnS. It can be concluded that Fe_3_O_4_@ZnS-SH had a better specific selectivity for Ag^+^, and this magnetic fluorescence nanoprobe could be used to detect Ag^+^. The inset picture presents TEM images of Fe_3_O_4_@ZnS-SH and that with the addition of Ag^+^, respectively. It can be seen clearly that Ag^+^ had been loaded and dispersed on the surface of Fe_3_O_4_@ZnS-SH magnetic fluorescent nanoparticles. The Fe_3_O_4_@ZnS-SH nanoparticles themselves had good dispersion and were almost spherical. The complexation of Ag^+^ ions and ethyl thioglycolate on the surface of Fe_3_O_4_@ZnS caused the aggregation of the probe, leading to distinct fluorescence quenching.

The effect of the concentration of ZnS on the fluorescence intensity of magnetic fluorescence nanoprobe Fe_3_O_4_@ZnS-SH was investigated. As Figure 9 shows, the fluorescence intensity at 425 nm was increased in the concentration range of 0–12.5 mM. However, the fluorescence intensity decreased as the concentration of ZnS continued to increase. Therefore, the concentration of ZnS was selected at 12.5 mM.

The pH value of the solution is one of the most important parameters affecting the detection ability of the probe. Especially in wastewater, the pH value will decrease with the increase in metal ion concentration. So, the effect of different pH values on the detection results was examined. As shown in Figure 10, the fluorescent intensity and quenching ratio almost did not change as the pH value of the solution varied from 4.0 to 9.0. Hence, the optimal pH value of the solution was chosen as 7.0.

The effect of different concentrations of Ag^+^ on the fluorescent intensity of the Fe_3_O_4_@ZnS-SH under optimized conditions was investigated. As shown in Figure 11a, the fluorescent intensity of the magnetic fluorescence nanoprobe decreased with the addition of increasing Ag^+^ concentration. Additionally, the fluorescent quenching ratio of the magnetic fluorescence nanoprobe linearly varied in the range of 0–400 µM (Figure 11b) with a low detection limit of 0.20 µM (3δ/k). The linear equation was I_0_/I = 0.01514x + 1.0740 (R^2^ = 0.9961). Compared with most of the methods reported before, the magnetic fluorescence nanoprobe Fe_3_O_4_@ZnS-SH had a wide linear range and a highly sensitive response to Ag^+^. The results indicate its superior analytical ability (Table 1).

According to the World Health Organization standard, the content of silver ions in drinking water should not be higher than 0.05 mg/L, and the concentration of silver ions in human body should be lower than 0.05 mg/L (0.46 μmol/L). The concentration value is just in the linear range, which confirms that the magnetic fluorescent nanoprobe determines the content of silver ions more accurately.

The fluorescence responses of the magnetic fluorescent nanoprobe Fe_3_O_4_@ZnS-SH to a series of interfering metal ions were recorded to evaluate the probe’s selectivity for Ag^+^ recognition (Figure 12a). As shown in Figure 12a, the interfering metal ions induced negligible ratio value changes with the increase in reaction time. However, the fluorescence intensity ratio was significantly increased when adding Ag^+^. These results indicate that the probe could be used in the detection of Ag^+^ specifically.

In addition, in the presence of other metal ions, the competitive selectivity of the magnetic fluorescent nanoprobe Fe_3_O_4_@ZnS-SH was also investigated. As shown in Figure 12b, the magnetic fluorescent nanoprobe Fe_3_O_4_@ZnS-SH could still respond to Ag^+^ with enhancing fluorescence ratio values, even in the presence of competitive metal ions. These results prove that the magnetic fluorescent nanoprobe Fe_3_O_4_@ZnS-SH showed high selectivity with Ag^+^ over other metal ions.

Finally, the fluorescence lifetimes of Fe_3_O_4_@ZnS, Fe_3_O_4_@ZnS-SH and Fe_3_O_4_@ZnS-SH with Ag^+^ were studied and compared (Figure 13).The fluorescence lifetime is sensitive to various internal and external factors defined by the fluorophore structure, including temperature, polarity and the presence of fluorescence quenchers As can be seen from Figure 13, these three samples presented similar exponential decay curves, and the average lifetimes of Fe_3_O_4_@ZnS, Fe_3_O_4_@ZnS-SH and Fe_3_O_4_@ZnS-SH/Ag^+^ systems were 2.79, 3.44 and 3.17 ns, respectively. The fluorescence lifetime measurements showed a much longer decay for Fe_3_O_4_@ZnS-SH than for Fe_3_O_4_@ZnS. The considerable improvement in the combination lifetime was primarily attributed to insertion of the ethyl thioglycolate moiety, which controlled the granularity and dispersion of ZnS and led to the longer decay. With the load of Ag^+^, the increases in the non-radiative energy loss competed with normal spontaneous emission processes, which shortened the lifetime of Fe_3_O_4_@ZnS-SH.

### 3.4. Removal of Ag^+^

Adsorption capacity is an important parameter used to evaluate nanomaterials, and the adsorption capacity of this magnetic fluorescence nanoprobe was investigated. The magnetic fluorescence nanoprobe (0.5 mg) was added to the solution of Ag^+^ (50 mL, 100 μM). After the full reaction of the magnetic fluorescence nanoprobe with Ag^+^, the magnetic fluorescence nanoprobe was separated using an external magnetic field and the concentration of Ag^+^ in the solution was determined with inductively coupled plasma mass spectrometry. The adsorption capacity of this the magnetic fluorescence nanoprobe was calculated using the following equation [47]: Q = [(C_0_ - C) × V] × m^−1^, where C_0_ and C are the initial and final concentrations of Ag^+^ in solution (mg L^−1^), V is the solution volume (L), and m is the weight of the magnetic fluorescence nanoprobe. The calculated maximum adsorption capacity was about 107.022 mg g^−1^ and the removal rate of the probe was 99.09%. The comparison of the adsorption capacity of Ag(I) with that of other adsorbents is presented in Table 2. The results show that the adsorption capacity of Fe_3_O_4_@ZnS-SH was superior to that of most of the adsorbents, suggesting that this magnetic fluorescence nanoprobe could became a potential efficient adsorbent for the removal of Ag(I).

### 3.5. Detection of Ag^+^ in Real Samples

To verify the feasibility of our magnetic fluorescent nanoprobe in complex samples, three samples were analyzed using the standard addition method. As shown in Table 3, the recoveries of three different concentrations of Ag^+^ ions ranged from 87.00% to 107.50%, with the relative standard deviation (RSD) being less than 4.5%. These results indicate that the magnetic fluorescent nanoprobe could be used for the determination of Ag^+^ ions in real samples with accuracy and reliability.

### 3.6. Detection Mechanism

To gain insights into the detection mechanism, FT–IR was performed to identify the specific binding sites between the magnetic fluorescent nanoprobe and Ag^+^. The result can be clearly observed in Figure 14, when the Fe_3_O_4_@ZnS-SH interacted with Ag+, the tensile peak of methoxy (C-O) was changed from 1287 cm^−1^ to 1372 cm^−1^, while the characteristic absorption peak of C=O moved from 1729 cm^−1^ to 1736 cm^−1^ [54]. Ethyl thioglycolate was chosen to enhance the coordination between the probe and Ag^+^, and the conclusion was supported by the experimental results.

## 4. Conclusions

In summary, ethyl thioglycolate-modified Fe_3_O_4_@ZnS@Ethyl thioglycolate nanoparticles were successfully synthesized and characterized. A new fluorescence detecting method based on the fluorescence quenching of the Fe_3_O_4_@ZnS@Ethyl thioglycolate nanoparticle in the presence of Ag^+^ was established to determinate Ag^+^ with high sensitivity and selectivity. Compared with other detection methods, this magnetic fluorescent nanoprobe provides easy operation and high accuracy and can be successfully used for Ag^+^ in practical samples. This magnetic fluorescent nanoprobe provides a new method for enriching, detecting and separating silver ions, which reduces the pollution of Ag^+^ and fluorescence probes in the environment. At the same time, it also provides a new promising platform for the detection of other heavy metal ions.

## Figures and Tables

**Figure 1 nanomaterials-13-01992-f001:**
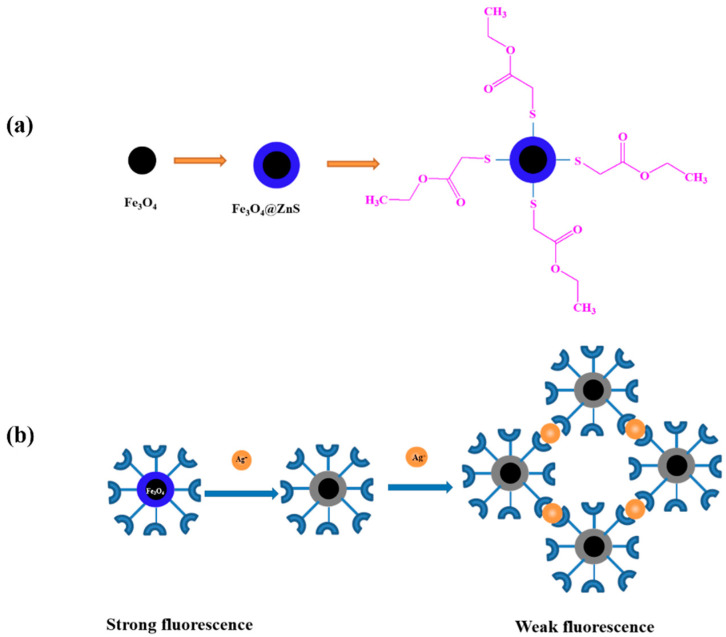
The structure of Fe_3_O_4_@ZnS@Ethyl thioglycolate magnetic nanoparticles (**a**) and the recognition property of magnetic fluorescence nanoprobe Fe_3_O_4_@ZnS-SH for Ag^+^ (**b**).

**Figure 2 nanomaterials-13-01992-f002:**
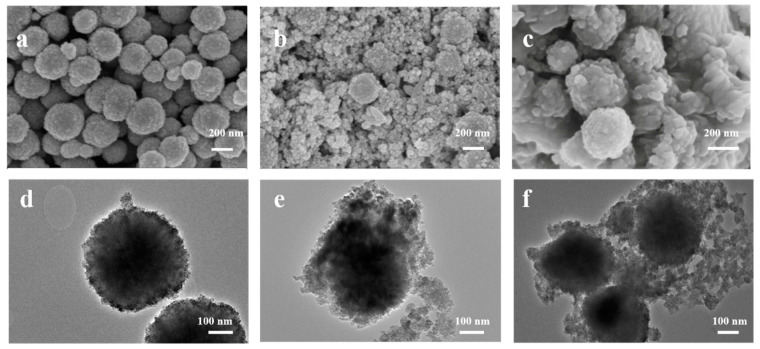
The SEM images of (**a**) Fe_3_O_4_, (**b**) Fe_3_O_4_@ZnS, and (**c**) Fe_3_O_4_@ZnS@Ethyl thioglycolate and TEM images of (**d**) Fe_3_O_4_, (**e**) Fe_3_O_4_@ZnS, and (**f**) Fe_3_O_4_@ZnS@Ethyl thioglycolate.

**Figure 3 nanomaterials-13-01992-f003:**
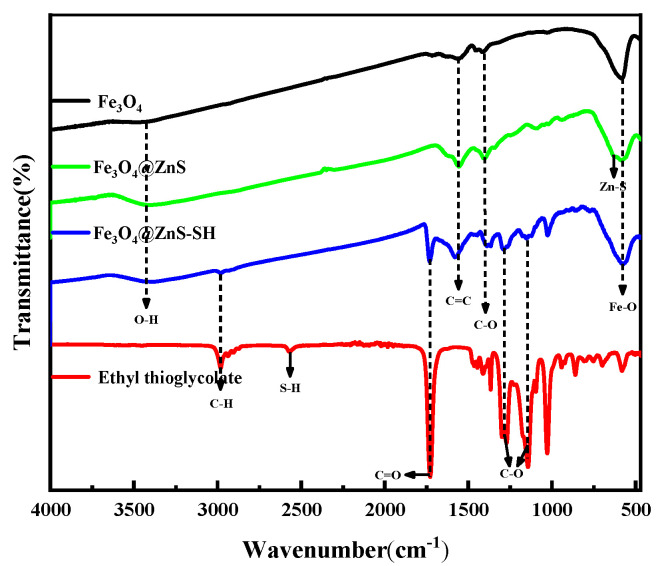
FT−IR spectra of Fe_3_O_4_, Fe_3_O_4_@ZnS, Fe_3_O_4_@ZnS@Ethyl thioglycolate and ethyl thioglycolate.

**Figure 4 nanomaterials-13-01992-f004:**
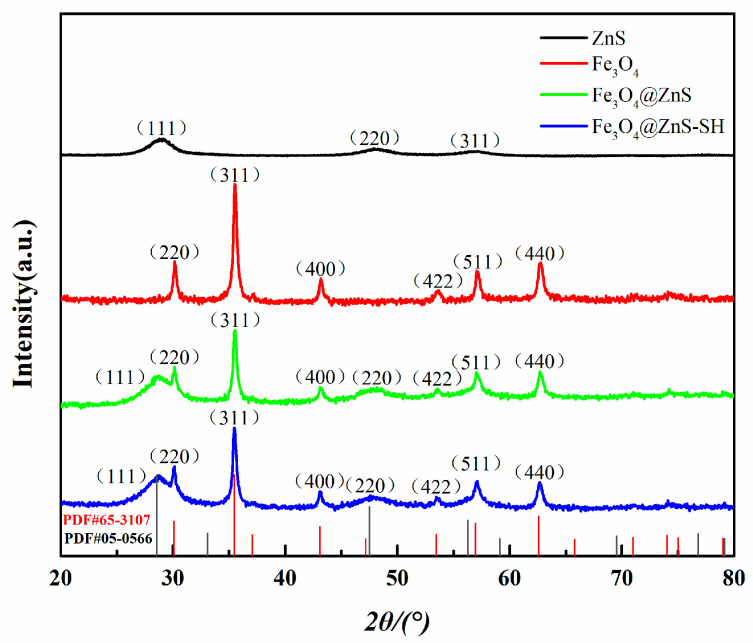
XRD patterns of ZnS, Fe_3_O_4_, Fe_3_O_4_@ZnS and Fe_3_O_4_@ZnS@Ethyl thioglycolate.

**Figure 5 nanomaterials-13-01992-f005:**
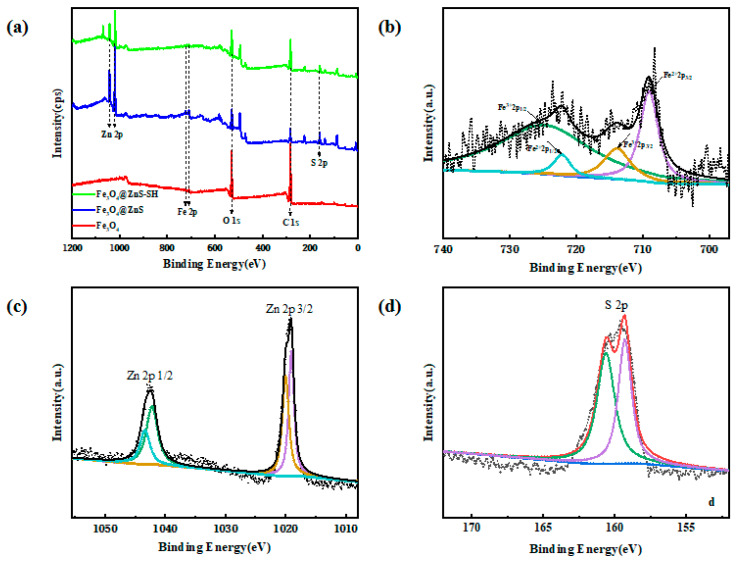
XPS analysis of Fe_3_O_4_, Fe_3_O_4_@ZnS and Fe_3_O_4_@ZnS@Ethyl thioglycolate (**a**). High-resolution XPS spectra of Fe2p (**b**), Zn2p (**c**), and S2p (**d**) of Fe_3_O_4_@ZnS@Ethyl thioglycolate.

**Figure 6 nanomaterials-13-01992-f006:**
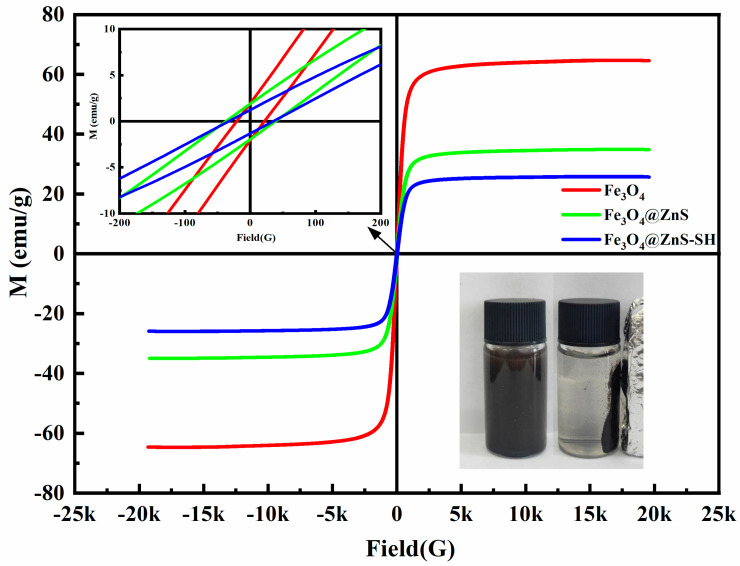
VSM measurements of Fe_3_O_4_, Fe_3_O_4_@ZnS and Fe_3_O_4_@ZnS@Ethyl thioglycolate.

**Figure 7 nanomaterials-13-01992-f007:**
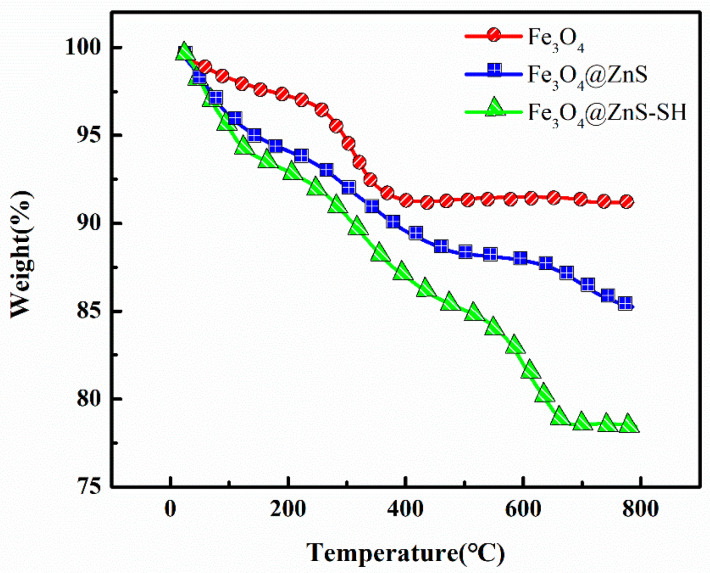
TGA curves of Fe_3_O_4_, Fe_3_O_4_@ZnS and Fe_3_O_4_@ZnS@ Ethyl thioglycolate.

**Figure 8 nanomaterials-13-01992-f008:**
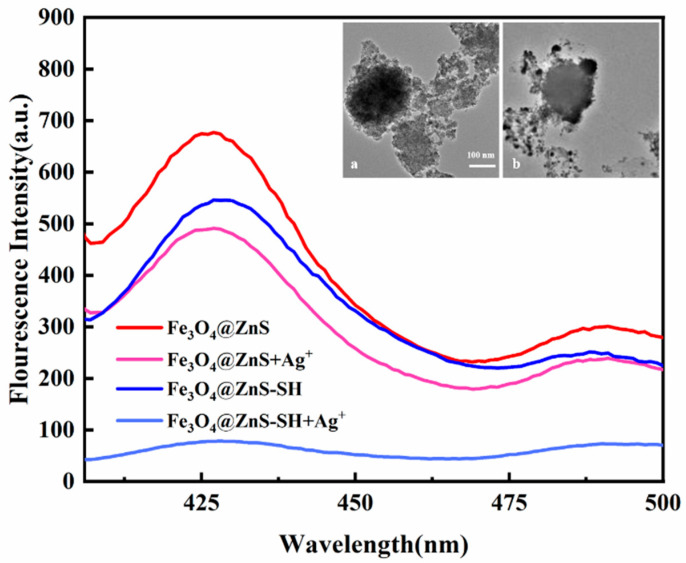
The comparison of the fluorescence spectra of Fe_3_O_4_@ZnS and Fe_3_O_4_@ZnS-SH before and after addition of Ag^+^. Inset: TEM images of Fe_3_O_4_@ZnS-SH (**a**) and with the addition of Ag^+^ (**b**).

**Figure 9 nanomaterials-13-01992-f009:**
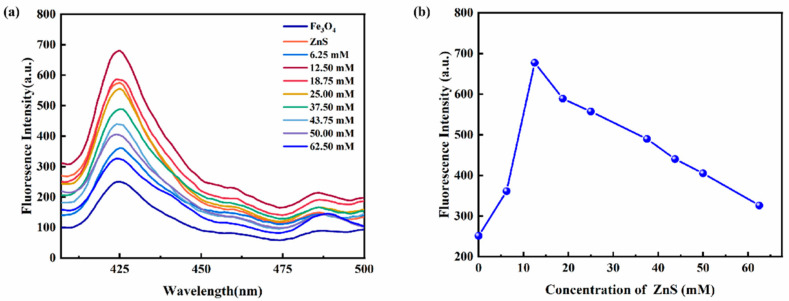
(**a**) The emission spectra of Fe_3_O_4_@ZnS at different concentrations of ZnS; (**b**) the effect of the ZnS concentration on the fluorescence intensity of Fe_3_O_4_@ZnS.

**Figure 10 nanomaterials-13-01992-f010:**
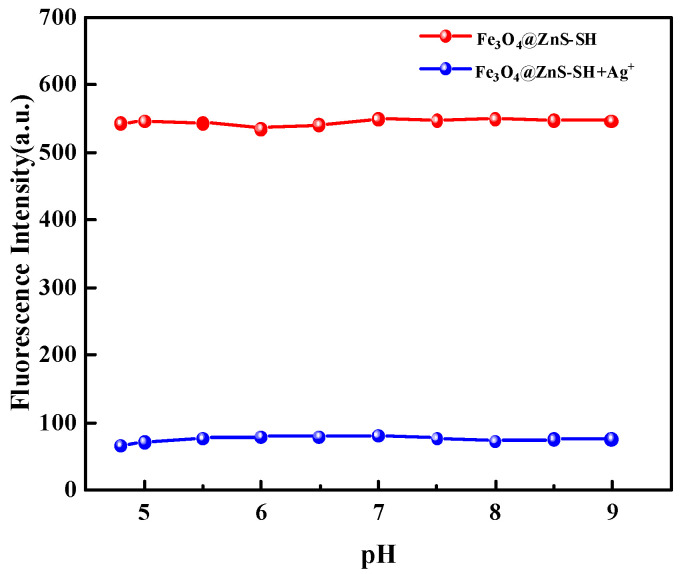
Fluorescence intensity of Fe_3_O_4_@ZnS-SH in the absence and presence of Ag^+^ at different pH values.

**Figure 11 nanomaterials-13-01992-f011:**
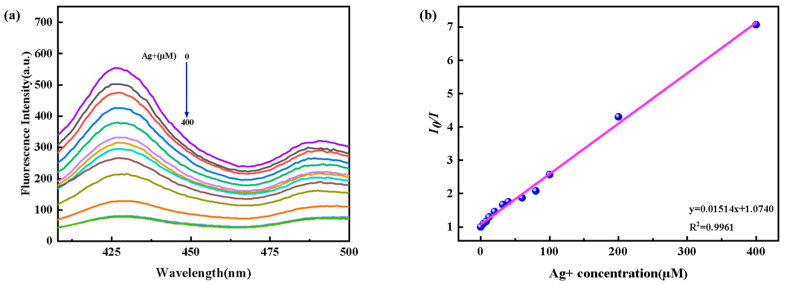
(**a**) Emission spectra of Fe_3_O_4_@ZnS-SH in the presence of increasing amounts of Ag^+^ at room temperature; (**b**) the curve of fluorescence intensity at 425 nm vs. Ag^+^ concentration. The concentrations of Ag^+^ are 0, 4, 8, 12, 20, 32, 40, 60, 80, 100, 200, and 400 μM, respectively.

**Figure 12 nanomaterials-13-01992-f012:**
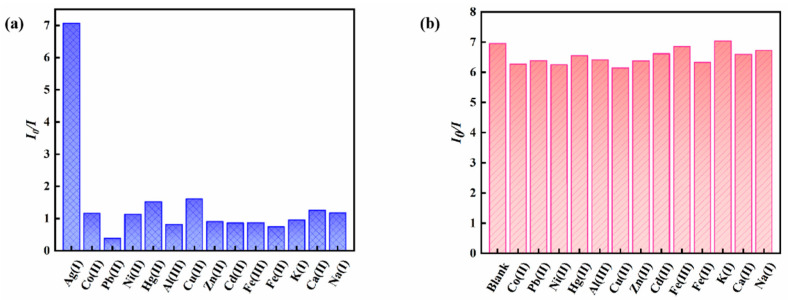
(**a**) The ratio of fluorescence quenching of Fe_3_O_4_@ZnS-SH in the presence of different metal ions; (**b**) the ratio of fluorescence quenching of Fe_3_O_4_@ZnS-SH upon the addition of Ag^+^ in the presence of other metal ions (Ag^+^, Co^2+^, Ni^2+^, Al^3+^, Cu^2+^, Zn^2+^, Cd^2+^, Fe^3+^, Fe^2+^, K^+^, Ca^2+^, Na^+^ (400 µM); Pb^2+^, Hg^2+^ (40 µM)).

**Figure 13 nanomaterials-13-01992-f013:**
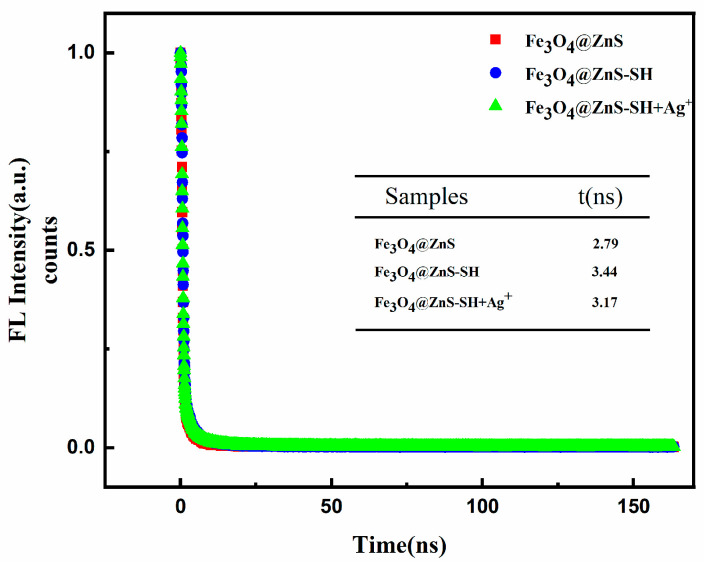
Fluorescence intensity decay curves of Fe_3_O_4_@ZnS, Fe_3_O_4_@ZnS-SH and Fe_3_O_4_@ZnS-SH with the addition of Ag^+^.

**Figure 14 nanomaterials-13-01992-f014:**
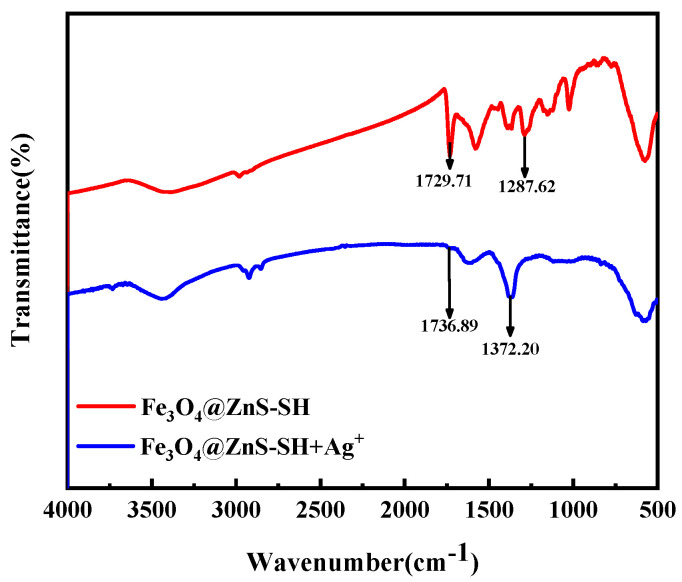
FT−IR spectra of Fe_3_O_4_@ZnS-SH before and after Ag(I) detection.

**Table 1 nanomaterials-13-01992-t001:** Comparison of the present nanoprobe with other reported sensors for Ag^+^ detection.

Type of Sensor	Detection Range	Detection Limit	Reference
N-doped Ti_3_C_2_ quantum	0.5–150 μM	0.1 μM	[3]
NPCl-doped carbon quantum dots	15.89–27.66 μM	26.38 μM	[5]
Nitrogen0doped carbon dots	0–40 μM	0.11 μM	[12]
Dual-element-doped carbon quantum dots	0.1–700 μM	0.03 μM	[31]
Acetaldehyde-modified g-C_3_N_4_ nanosheets	0–30 μM	1.0 μM	[46]
Fe_3_O_4_@ZnS-SH	0–400 μM	0.20 μM	This work

**Table 2 nanomaterials-13-01992-t002:** Comparison with other reported adsorbents for Ag(I).

Adsorbents	Maximum Adsorption Capacities (mg/g)	Ref.
Fe_3_O_4_/polypyrrole nanocomposite	143.3	[48]
Activated carbon/γ-Fe_2_O_3_- 4,4′-bis-(3-phenylthiourea)diphenyl methane	32.6	[49]
Fe_3_O_4_@SiO_2_@Tio_2_- Ion imprinting polymers	35.48	[50]
Fe_3_O_4_@SiO2-HO-S	166.32	[51]
Fe_3_O_4_-sulfur-functionalized polyamidoamine	139.32	[52]
Fe_3_O_4_@microbial extracellular polymeric substance	47.6	[53]
Fe_3_O_4_@ZnS-SH	107.022	This work

**Table 3 nanomaterials-13-01992-t003:** Determination of Ag^+^ in real samples.

Sample	Added (μM)	Measured (μM)	Recovery (%)	RSD (%)
Tap water	1 5.0 10.0	0.98 5.29 10.30	98.00 105.80 103.00	2.3 3.1 2.9
Mineral water	1 5.0 10.0	1.02 4.96 10.75	102.00 99.20 107.50	3.2 2.7 2.5
Electrolysis wastewater	1 5.0 10.0	0.87 4.43 8.75	87.00 88.60 87.50	4.5 3.9 4.0

## Data Availability

The data presented in this study are available upon request from the corresponding author.

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
