# Peer review of "An Ethyl-Thioglycolate-Functionalized Fe3O4@ZnS Magnetic Fluorescent Nanoprobe for the Detection of Ag+ and Its Applications in Real Water Solutions"

_nanomaterials, 2023, doi:10.3390/nano13131992_

Round 1
Reviewer 1 Report
The Manuscript reports the preparation of a new Fe3O4@ZnS magnetic nanoprobe functionalized with ethyl thioglycolate (Fe3O4@ZnS-SH) for detection Ag+. The probe, as well as its precursors, Fe3O4 and Fe3O4@ZnS, were characterized by SEM and TEM, XRD, XPS, IR, magnetic techniques, and thermogravimetry. The fluorescence response of the probes to Ag+ ions in the presence of other analytes was studied. The results show a good turn-off sensitivity of Fe3O4@ZnS-SH towards Ag+ with a detection limit of 0.2 μM, which is superior to previously reported probes. In addition, a good adsorption capacity of Fe3O4@ZnS-SH was shown, removing up to 99.09% Ag+. Despite its impressive scientific significance, the Manuscript suffers from a number of inaccuracies and inconsistencies and requires major revision.
1. In the Introduction, the Authors refer to the studies of Fe3O4 nanoparticles, but they do not mention recent studies of Fe3O4@ZnS at all. In fact, there are a number of works devoted to Fe3O4@ZnS nanocomposites, e.g. “Fabrication of Fluorescent Magnetic Fe3O4@ZnS Nanocomposites” doi: 10.1166/jnn.2014.8239; “Fabrication of Fe3O4/ZnS nanocomposites towards ultrasensitive resonant Raman scattering-based immunoassays” doi: 10.1016/j.matlet.2019.06.092; “Synthesis of Magnetic MWNTs/ZnS/Fe3O4 Nanocomposites and Their Enhanced Photocatalytic Activity” doi: 10.1080/15533174.2015.1031047; “Synthesis and Characterization of ZnS@Fe3O4 Fluorescent-Magnetic Bifunctional Nanospheres” doi: 10.1016/j.spmi.2017.08.044, etc. Please conduct a more in-depth analysis of the literature and describe the current state of the art in this area.
2. The detection mechanism (Section 3.6) and representation of Ag+ binding in Fig. 1 are questionable. The COOR group of ethers is known to have a very poor binding activity to metal ions, especially in the presence of the sulphide groups of ZnS (Ag+ has a very high affinity for S2–). In addition, the vibrations of C-O and C=O groups of ethyl thioglycolate should not disappear upon coordination of a metal. Thus, I would recommend revising the conclusion about the binding mechanism. Disappearance of the C-O and C=O vibrations can indicate that Fe3O4@ZnS-SH dissociates upon addition of Ag+ with splitting off ethyl thioglycolate and the formation of some transformed particles, for instance, Fe3O4@ZnS-Ag.
3. The presence or absence of Ag+ in the adsorption product is indeed an interesting question. Probably, Ag+ is reduced to metallic Ag0 by organic species (ethyl thioglycolate) and precipitates from the solution. Could the Authors comment on this pathway?
4. The IR spectroscopy and XPS data for Fe3O4 reveal the presence of significant amount of carbon, which normally should not be present in Fe3O4. The Authors should discuss the origin of carbon (Fe3O4@acetate?) in the text. What is the role of CH3COONa in the synthesis? How it affects the properties of Fe3O4@ZnS-SH? Why do not the Authors use a more general pathway to Fe3O4 without the acetate? Please discuss these points in the text.
5. In the synthesis of Fe3O4@ZnS, ammonia was used. What is the role of ammonia? Can NH3 be present in the product (Fe3O4@ZnS-NH3)? How the Authors prove its presence/absence in the product?
6. Please specify the amount of Fe3O4 and Fe3O4@ZnS (in mg or mg/L) in the synthetic procedures 2.3 and 2.4, correspondingly.
7. In the synthesis of Fe3O4@ZnS-SH, NaOH was used. What is its role? Please specify the amount of NaOH (in mg or mg/L) in the synthetic procedure 2.4.
8. Page 5, line 154: “As to the spectra of Fe3O4@ZnS, the absorption peaks at 580 cm-1 and 1018 cm-1 representing Fe-O and Zn-S, respectively” The Zn-S vibrations normally locate at ca. 500 cm-1, but not at 1000 cm-1. Probably, the high-frequency bands are associated with the acetate. Please revise the sentence.
9. Abstract: the term “good water solubility” is not appropriate in this case, since the probe is actually not soluble. Please revise.
Reviewer 2 Report
In this work, the authors have studied the synthesis and performance of a functionalized Fe3O3@ZnS nanoprobe for the detection and removal of Ag+ from aqueous media. The paper is interesting and novel. However, it needs to be major-revised before being reconsidered for possible publication in Nanomaterials.
1.The manuscript was incomplete. As such, I couldn’t evaluate the whole work. The authors moved some of their results into a supplementary file; however, the file couldn’t be downloaded from the susy system. Only the main manuscript file was present. It is necessary to present all the results in one file. The publisher does not pose any restriction on the number of pages. Therefore, all results should be collected and presented in the main manuscript file.
2.Fig. 1b is a schematic representation of the recognition property of the prepared composite. However, the symbols are not explained. You need to explain the different symbols. Why is the ZnS coating suddenly gray instead of blue? Why is the Ag+ symbol missing in the right figure?
3.You need to sufficiently explain the specific recognition of Ag+ among different transition metal cations. Why is this probe specific for Ag+ cations only?
4.JCPDS card numbers should be included in the legend of Fig. 3.
5.Magnetic hysteresis loops are missing in the manuscript. The magnetization saturation of Fe3O4 nanoparticles is mentioned (63 emu/g); however, it should be compared with previous reports (https://doi.org/10.3390/nano11061614, https://doi.org/10.3390/nano12111786, etc.)
6.The insets in Fig. 5 must include scale bars.
7.Adsorption curves are missing in the manuscript. Only the maximum adsorption capacities are presented (Table 2). Which isotherm was used to fit the data?
8.Abbreviations used in Table 2 (PPy, BPDM, IIP, etc.) must be defined in the footnote.
9.The difference in the chemical composition of mineral and tap water should be mentioned (Table 3).
10.Tables are not properly numbered. Table 1 on p. 9 cannot come after Table 3 (p. 8).
11.References in the right column of Table 1 (p. 9) should be given by numbers and collected in the reference section.
12.All abbreviations, even those that are common, e.g., WHO (world health organization), LOD (lowest detection limit), must be explained at their first mention in the manuscript.
13.The selectivity of the probe towards the Ag+ cation (Fig. 6a) must be sufficiently discussed. The discussion of the mechanism is very short (section 3.6). Furthermore, the results should be compared with previously used probes.
Round 2
Reviewer 1 Report
The Authors have paid little attention to the comments, so the Manuscript still needs revision.
1. Point 2 (about C-O vibrations). The answer was: ”The vibration of C-O and C=O of ethyl thioglycolate at (1157 cm-1, 1288 cm-1; 1731 cm-1) in FT-IR spectrum disappeared, which might be due to the coordination between ethyl thioglycolate and Ag+. Compared with the known literature, the results can be confirmed (https://doi.org/10.1016/j.microc.2023.108633).”
Unfortunately, the work https://doi.org/10.1016/j.microc.2023.108633 contains no evidence for the disappearance of the vibrations upon coordination, just a simple unproven statement. It is well known that C-O and C=O vibrations are clearly distinguishable in IR spectra and do not disappear, but shift upon coordination. This is described in many books and guides, e.g. by Nakamoto, Infrared and Raman Spectra of Inorganic and Coordination Compounds, and is observed in many Ag complexes. Here are some randomly taken examples: https://doi.org/10.1080/00958972.2014.1003051 https://doi.org/10.22034/ajca.2022.322123.1294 https://doi.org/10.3390/ijms13066639 . Thus, I insist on further revision of this point as well as point 3.
2. Point 4 (about the presence of carbon in Fe3O4). The answer was: “In the initial stage of the reaction, HOCH2CH2OH as a dispersion medium provides a liquid environment for the dissolution of FeCl3·6H2O and CH3COONa, CH3COONa water releases OH-, and the alkaline environment accelerates the hydrolysis of FeCl3·6H2O. In the middle stage of the reaction, Fe3+ and OH- reacted to form an intermediate phase Fe(OH)3, and part of Fe3+ was reduced to Fe2+ by HOCH2CH2OH in the liquid phase system. In the later stage of the reaction, Fe(OH)3 combined with Fe(OH)2 to form Fe3O4”
The main part of the comment was left unanswered. The Authors should discuss in the Manuscript the presence of significant amount of carbon, which normally should not be present in Fe3O4.
3. Point 5 (about the amount of Fe3O4 and Fe3O4@ZnS). The answer was: “In the synthesis of Fe3O4, only the amount of Fe that added to the solution but not the concentration can be specific to determine. So, a fixed volume is taken each time.”.
I mean the syntheses of Fe3O4@ZnS and Fe3O4@ZnS-SH (synthetic procedures 2.3 and 2.4, correspondingly). Please specify (in grams) the loadings of Fe3O4 and Fe3O4@ZnS used for the syntheses.
4. Point 8 (about the vibrations at 1000 cm–1). The answer was: “As to the spectra of Fe3O4@ZnS, the absorption peaks at 580 cm-1and 1018 cm-1 representing Fe-O and Zn-S, respectively. Compared with the known literature, the results can be confirmed (https://doi.org/10.1016/j.msec.2018.12.147).”
Unfortunately, the work https://doi.org/10.1016/j.msec.2018.12.147 contains no evidence for the presence of Zn-S vibrations at high frequencies, just a simple unproven statement. It is well known that metal-to-sulphur vibrations arise at low frequencies at 500 cm–1 and lower. This is described in many books and guides, e.g. by Nakamoto, Infrared and Raman Spectra of Inorganic and Coordination Compounds. The presence of high-frequency vibrations is normally associated with O-H or C-C/C-O or S-O groups that present on the surface of nanoparticles. Here are some randomly taken works: https://dx.doi.org/10.1021/acs.jpcc.9b11323 https://doi.org/10.22052/JNS.2018.02.001 . Thus, I insist on further revision of this point.
Reviewer 2 Report
The authors sufficiently answered most of my comments. The paper is interesting and worthy of publication as it contains many important experimental results. It can be published subject to minor revision:
Please, strengthen the discussion part, especially the section 3.6 as it has 7 lines only (lines 347-353). It is possible that the vibration bands of ethylthioglycolate have not completely disappeared but were only shifted due to coordination. You should provide a detail of the IR spectra at 2000 – 500 cm-1. You should assign and discuss the vibration bands sufficiently by referring to previous literature on inorganic complexes, e.g., https://doi.org/10.3390/ijms13066639. If this point is addressed, the paper can be accepted for publication.
Round 3
Reviewer 1 Report
The Authors have responded to all the points, and the Manuscript can be accepted for publication.